# Flavonoid Metabolome-Based Active Ingredient Mining and Callus Induction in *Catalpa bungei* C. A. Mey

**Xiaofeng Zeng** [1,2,†]**, Xiao Wang** [1,2,†]**, Yanling Zeng** [1,2,*]**, Jinbo Hou** [1,3] **and Zhiming Liu** [4,*]

[1] Key Laboratory of Cultivation and Protection for Non-Wood Forest Trees of Ministry of Education, Central South University of Forestry and Technology, Changsha 410004, China; t20032076@csuft.edu.cn (X.Z.); xlh1302@163.com (X.W.); 13731815383@126.com (J.H.)

[2] Key Laboratory of Non-Wood Forest Products of State Forestry Administration, Central South University of Forestry and Technology, Changsha 410004, China

[3] Anhui Hongsen Hi Tech Forestry Co., Ltd., Bozhou 236814, China

[4] Department of Biology, Eastern New Mexico University, Portales, NM 88130, USA

[*] Correspondence: t20061125@csuft.edu.cn (Y.Z.); zhiming.liu@enmu.edu (Z.L.)

[†] These authors contributed equally to this work.

**Abstract:** *Catalpa bungei* C. A. Mey is a unique and precious multi-purpose tree species that possesses great timber-related, ornamental and medicinal values. In this study, MS, N6 and DKW were used as basic media, and different concentrations of 6-BA and NAA were added for callus induction. The induction rate and total flavonoid content of callus tissue showed that the best callus induction medium was DKW + 2.0 mg·L$^{-1}$ 6-BA + 0.5–1.0 mg·L$^{-1}$ NAA. The leaves of different bark phenotypes of *C. bungei* C. A. Mey and the callus tissue extracted from young leaves of *C. bungei* C. A. Mey were used as experimental materials to construct metabolomic profiles of widely targeted flavonoids. Based on the metabolomic databases, the predominant flavonoids were screened from the callus tissues. Eight flavonoid metabolites increased in callus, and diosmetin-7-O-rutinoside (diosmin) was the flavonoid constituent with the shortest retention time, most efficient detection and best medicinal functions among these 8. The optimal medium for callus induction was supplemented with different concentrations of elicitors (salicylic acid SA and yeast extract YE). The optimal elicitor and the amount to be added were determined by analyzing the induction rate of callus, as well as the total contents of flavonoids and diosmin. The addition of SA and YE in appropriate amounts increased the total flavonoid content in the callus, but only the addition of YE promoted the formation of diosmin in the callus. The optimal medium formulation to promote the formation of callus was DKW + 2.0 mg·L$^{-1}$ 6-BA + 1.0 mg·L$^{-1}$ NAA + 30 g·L$^{-1}$ sucrose + 6.8 g·L$^{-1}$ agar + 10 μmol·L$^{-1}$ SA. The medium formulation to optimally increase the content of geraniol glucoside was DKW + 2.0 mg·L$^{-1}$ 6-BA + 1.0 mg·L$^{-1}$ NAA + 30 g·L$^{-1}$ sucrose + 6.8 g·L$^{-1}$ agar + 200 mg·L$^{-1}$ YE. The results of the present study will provide a scientific basis for the subsequent increase in the content of the active components of the suspension cells via the addition of elicitors, and for the production of diosmin in factory settings.

**Keywords:** *Catalpa bungei* C. A. Mey; callus induction; flavonoid metabolite; yeast extract; diosmetin-7-O-rutinoside (diosmin)





## 1. Introduction

*Catalpa bungei* C. A. Mey is a deciduous tree in the family Ziweiidae. It has high industrial value, ornamental value and medicinal value. It has been known as the "king of wood" since ancient times. In recent years, the research into the active components of *C. bungei* C. A. Mey has received more and more attention. The active components of *Catalpa bungei* C. A. Mey play an important role in anti-inflammatory, antibacterial, and antiviral qualities. Xu et al. [1] extracted ursolic acid from the leaves of *C. bungei* C. A. Mey, which were proven to inhibit the survival of cervical cancer cells. Fruits and seeds

of *Catalpa bungei* C. A. Mey contain nicotinic acid and citric acid that can be extracted as diuretics. They are good remedies for the treatment of exophthalmos, nephropathy, and wet peritonitis. Tang et al. [2] isolated two new chlorinated cycloenyl ether terpenoids from *C. bungei* C. A. Mey seeds and completed the characterization of 20 known compounds, demonstrating that the chemical constituents in *C. bungei* C. A. Mey seeds have inhibitory effects on the activities of soluble epoxide hydrolase, cholinesterase, and nuclear factor KB.

Zhuo et al. [3] extracted flavonoids from *C. bungei* C. A. Mey leaves and performed a comparative study to clarify the differences in the growth of this species and flavonoid content between different cultivation modes and different asexual lines. Xu et al. [4] determined that the total flavonoid content in *C. bungei* C. A. Mey leaves could reach 40.45 mg/g, and isolated two major flavonoid compounds in *C. bungei* C. A. Mey leaves, namely lignans and apigenin, which are antioxidants and have bacteriostatic activity.

Natural products are an important source of anticancer qualities and infectious disease drugs and have an irreplaceable role in medicine. A considerable portion of the natural products belong to plant secondary metabolites, but their contents in plants are very low, and their direct extraction from plants is costly because it is requires researchers to grow a large number of plants [5]. Plant cell culture provides an achievable basis for the large-scale production of secondary metabolites. Under suitable in vitro culture conditions, some secondary metabolites can be accumulated at a higher quantity in plant cultures than in whole plants, suggesting that the use of plant cell culture instead of whole plants for the production of specific secondary metabolites is promising [6]. Currently, *C. bungei* C. A. Mey cell engineering research is focused on the establishment of an in vitro fast propagation system [7] and the study of somatic cell embryogenesis [8]. Gao et al. [9] used immature embryos of *C. bungei* C. A. Mey to obtain embryonic calluses. These which were inoculated into 1/2MS suspension culture medium for oscillatory culture, and the researchers established a cell suspension line with good dispersion, fast proliferation, and high stability. Elicitors are a special class of active substances with strong specificity that can induce the expression of specific genes in plants in a targeted manner to promote cell synthesis and the accumulation of specific metabolites. According to their sources, two categories are usually recognized, these being abiotic and the biotic elicitors. Biotic elicitors mainly include microorganisms such as fungi, bacteria and their extracts, such as polysaccharides, salicylic acid, jasmonic acid, mannitol, abscisic acid, etc. Abiotic elicitors have features including high or low temperatures, ultraviolet rays, pH, heavy metal salts, highly concentrated salts, and ethylene etc [10]. There have been more reports of increasing the content of secondary metabolites in cells by adding biotic elicitors. However, studies on the addition of exogenous substances to increase the content of active substances in the callus have not been reported.

In this study, we analyzed the components with high utilization values based on the establishment of a *C. bungei* C. A. Mey flavonoid metabolome, established a *C. bungei* C. A. Mey callus induction system, and increased the content of flavonoids active components in *C. bungei* C. A. Mey callus by adding elicitors to the culture medium. The results from our study provides a scientific basis for the large-scale production of flavonoids through *C. bungei* C. A. Mey cell culture in the future.

## 2. Materials and Methods

### 2.1. Plant Material and Reagents

Three types of materials were used for the construction of the flavonoid metabolome database. (1) The calluses of 'Hongsen' *C. bungei* C. A. Mey were cultivated on a 2.3 optimum callus induction medium for 28 d, denoted as YS (Figure 1b); (2) on the leaves of 'Hongsen' *C. bungei* C. A. Mey with light-barked phenotypes, denoted as GP (Figure 1a); and (3) on the leaves of 'Hongsen' *C. bungei* C. A. Mey with wrinkled-barked phenotypes denoted as ZP (Figure 1a). Calluses was induced on young leaves of 'Hongsen' *C. bungei* C. A. Mey (Figure 1c). The materials were obtained from the base of 'Hongsen' *C. bungei* C. A. Mey in Eddy County, Bozhou City, Anhui Province, China.

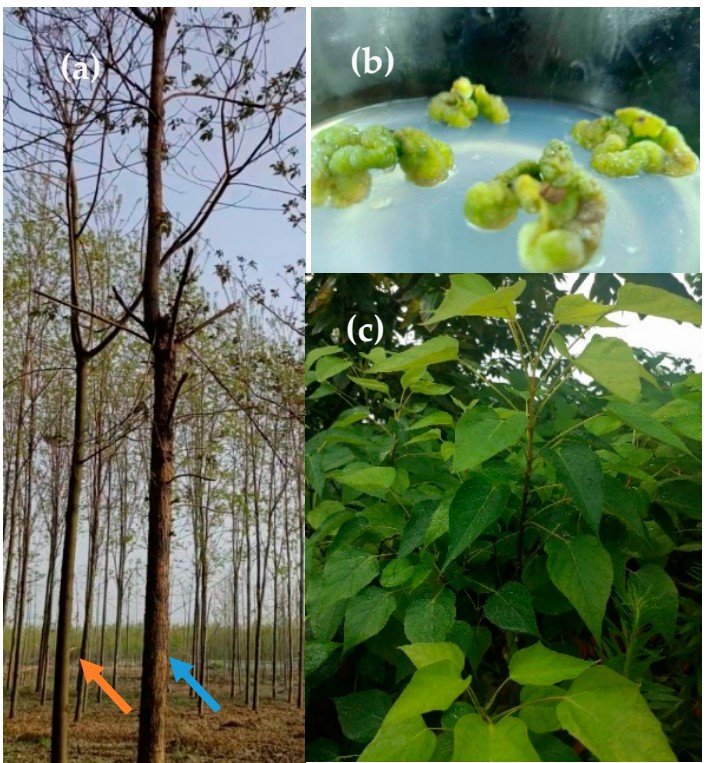

**Figure 1.** Materials used in this study: (**a**) Yellow arrow indicates light-barked *C. bungei* C. A. Mey, blue arrow indicates wrinkled-barked *C. bungei* C. A. Mey; (**b**) callus; (**c**) *C. bungei* C. A. Mey material for callus induction.

The components of medium: Murashige and Skoog culture medium (MS) (M8521), Chu's N-6 medium (N6) (LA6912), Driver and Kuniyuki and McGranahan medium (DKW) (LB0210), 6-benzylaminopurine (6-BA) (A8170), α-naphthalene acetic acid (NAA) (N8010), salicylic acid (SA) (S7080), and yeast extract (YE) (Y8020) from Solarbio (Beijing, China).

The standard: Rutin (B20771) and diosmin (B20287) from Yuanye Bio-Technology (Shanghai, China).

*2.2. Sterilization of Leaves of Catalpa bungei C. A. Mey*

The leaves were harvested at noon on a sunny day. They were placed in a beaker and soaked in washing powder solution for about 5 min with gentle stirring, and then rinsed for 30–40 min under tap water. After rinsing, a small amount of sterile water was added to keep the samples moist. Then, on the ultra-clean bench, the leaves were immersed in 75% alcohol for 15 s and 30 s. After rinsing them with sterile water 3–4 times, the leaves were sterilized with 0.1% mercuric chloride by immersion for 4 min, 6 min and 8 min, respectively, the specific treatment combinations were shown in Table 1. The bottles were shaken slightly and rinsed with sterile water 4–5 times. After sterilization, the leaves were placed on sterile filter paper, the leaf veins were excised using an aseptic blade along the midline, and the leaf blades were sliced into 0.5 cm × 0.5 cm squares. Each treatment group was inoculated using 10 bottles with 3 leaf squares per bottle in triplicate. The tissue culture materials were placed under an LED light source. The light intensity (PPFD) was $50 \pm 5$ μmol·m$^{-2}$·s$^{-1}$, the light time was 16 h/d, the temperature was $25 \pm 5$ °C, and the contamination, browning and survival were counted after 28 d of inoculation.

**Table 1.** Leaf disinfection situation.

| Group | Sterilization Methods | Contamination Rate (%) | Browning Rate (%) | Survival Rate (%) |
|---|---|---|---|---|
| 1 | 75% alcohol 15 s + 0.1% mercuric chloride 2 min | 17.78 ± 4.01 a | 10 ± 3.85 c | 72.22 ± 1.11 a |
| 2 | 75% alcohol 15 s + 0.1% mercuric chloride 4 min | 8.89 ± 2.93 b | 11.11 ± 2.94 c | 80 ± 5.09 a |
| 3 | 75% alcohol 30 s + 0.1% mercuric chloride 2 min | 4.44 ± 4.44 b | 33.34 ± 3.33 b | 62.22 ± 6.76 ab |
| 4 | 75% alcohol 30 s + 0.1% mercuric chloride 4 min | 0 b | 16.67 ± 10.72 bc | 83.33 ± 10.72 a |
| 5 | 75% alcohol 30 s + 0.1% mercuric chloride 6 min | 0 b | 34.44 ± 5.88 b | 65.56 ± 5.88 a |
| 6 | 75% alcohol 30 s + 0.1% mercuric chloride 8 min | 0 b | 58.89 ± 7.78 a | 41.11 ± 7.78 b |

Means followed by the same letters in rows are not significantly different at $p \leq 0.05$.

### 2.3. Callus Induction in Leaves of Catalpa bungei C. A. Mey

As shown in Table 2, MS, N6 and DKW were used as the basic media, and different concentrations of 6-BA and NAA were added. 6-BA was set at concentrations of 0.5 mg·L$^{-1}$, 1.0 mg·L$^{-1}$ and 2.0 mg·L$^{-1}$, NAA was set at concentrations of 0.5 mg·L$^{-1}$, 1.0 mg·L$^{-1}$ and 1.5 mg·L$^{-1}$, and the experiments were conducted in an orthogonal design using the L$_9$(3$^3$). The medium was supplemented with sucrose 30 g·L$^{-1}$ and agar 6.8 g·L$^{-1}$, and pH was adjusted to 5.8–5.9. Leaves treated using the optimal disinfection method were inoculated into each treatment group. Each treatment group was inoculated with 10 bottles of 3 leaves per bottle, repeated 3 times. The tissue culture materials were placed under an LED light source. The light intensity (PPFD) was 50 ± 5 μmol·m$^{-2}$·s$^{-1}$, the light time was 16 h/d, the temperature was 25 ± 5 °C, and the induction rate and total flavonoid content were counted after 28 d of inoculation.

**Table 2.** Callus induction of leaf.

| Group | Hormone Concentration (mg·L$^{-1}$) 6-BA | NAA | Basic Medium | Induction Rate (%) | Total Flavonoid Content (mg·g$^{-1}$) |
|---|---|---|---|---|---|
| A | 0.5 | 0.5 | MS | 52.78 ± 2.78 Bb | 66.50 ± 0.55 e |
| B | 1.0 | 1.5 | MS | 61.11 ± 2.78 ABab | 77.67 ± 1.34 d |
| C | 2.0 | 1.0 | MS | 66.67 ± 4.82 ABa | 77.29 ± 2.46 d |
| D | 0.5 | 1.5 | N6 | 0 Cc | 0 g |
| E | 1.0 | 1.0 | N6 | 2.78 ± 2.78 Cc | 0 g |
| F | 2.0 | 0.5 | N6 | 8.33 ± 4.81 Cc | 46.21 ± 0.41 f |
| G | 0.5 | 1.0 | DKW | 63.89 ± 2.78 ABa | 108.71 ± 1.04 a |
| H | 1.0 | 0.5 | DKW | 69.45 ± 2.78 Aa | 86.55 ± 1.37 c |
| I | 2.0 | 1.5 | DKW | 66.67 ± 4.81 ABa | 98.34 ± 1.15 b |
| K1 | 1.1667 | 1.3056 | 1.8056 | | |
| K2 | 1.3343 | 1.3334 | 0.1111 | | |
| K3 | 1.4167 | 1.2778 | 2.0001 | | |
| R1 | 0.2500 | 0.0556 | 1.8889 | | |
| K4 | 175.21 | 199.26 | 221.46 | | |
| K5 | 164.22 | 186.00 | 46.21 | | |
| K6 | 221.84 | 176.01 | 293.60 | | |
| R2 | 57.62 | 23.25 | 247.39 | | |

Means followed by the same letters in rows are not significantly different at $p \leq 0.05$.

### 2.4. Establishment of Flavonoids Widely Target Metabolomics from Catalpa bungei C. A. Mey

The metabolome database was built by Wuhan Mai Tver Biotechnology Co., Ltd. Chromatographic conditions were as follows: an Agilent SB-C18 (2.1 mm × 100 mm, 1.8 μm) column was used. The mobile phases were 0.1% formic acid aqueous solution (A) and acetonitrile (0.1% formic acid was added to B). The elution procedure was 5% B at

0 min, the B-phase was increased to 95% after 9 min and this was maintained for 1 min, and then this was decreased to 5% after 10–11 min and maintained for 3 min. The flow rate was 0.35 mL/min. The column temperature was 40 °C, and the injection volume was 4 μL. The mass spectrometry conditions were as follows: the ion source was an electrospray ionization source (ESI) at 550 °C. Positive and negative ions were detected with a spray voltage (IS) of 5500 V (for positive ions)/−4500 V (for negative ions). The pressure of the ion source gas I (GSI) was 50 psi, and that of the ion source gas II (GSII) was 60 psi. Curtain gas (CUR) pressure was 25 psi, and the collision-induced ionization parameter was set to high and MRM mode quantification. The de-cluster voltage (DP) and collision energy (CE) were optimized and the DP and CE of the ion pairs were monitored.

### 2.5. Elicitor Selection for Flavonoids Accumucation in Catalpa bungei C. A. Mey Leaf Callus

The optimum callus induction medium for *C. bungei* C. A. Mey leaves was obtained using 2.3, to which different concentrations of salicylic acid (SA: 10 μmol·L$^{-1}$, 50 μmol·L$^{-1}$, 100 μmol·L$^{-1}$) and yeast extracts (YE: 50 mg·L$^{-1}$, 100 mg·L$^{-1}$, 200 mg·L$^{-1}$) were added (Table 2). Each group of treatments was inoculated with 10–12 explants. This was replicated three times and placed under the LED white light source for cultivation, with a photon passage factor (PPFD) of 50 ± 5 μmol·m$^{-2}$·s$^{-1}$, a light time of 16 h/d and a temperature of 25 ± 2 °C. The induction rate, total flavonoid content and geranylgeranyl glycoside content were assessed 28 d after inoculation. The optimum elicitor and its concentration were determined on the basis of total flavonoid and diosmin content in the callus tissues.

### 2.6. Determination of Total Flavonoid Content of Catalpa bungei C. A. Mey Callus

The calluses were oven-dried at 60 °C until a constant weight was reached. Then, they were ground into powder and passed through a 60-mesh sieve. In total, 0.2 g of the powders was added into a conical flask containing 70% alcohol according to 1:100 material–liquid ratio and then extracted for 60 min using ultrasound. After that, the extracting solution was filtered and set into a 50 mL volumetric flask on standby.

Standard solution preparation: rutin standards (5 mg) were added into a small amount of 70% ethanol. After dissolution, the solution was transferred into a 25 mL volumetric flask then diluted with 70% ethanol to scale and shaken well to form the standard solution (0.2 mg·mL$^{-1}$). Ultraviolet spectrophotometer detection (UV-1900i, Shimadzu, Japan): the rutin standard solutions were absorbed at volumes of 0 mL, 1 mL, 2 mL, 3 mL, 4 mL, and 5 mL precisely, placed into six 20 mL stoppered test tubes, and sequentially numbered 0, 1, 2, 3, 4, and 5 in each test tube. We added 5% sodium nitrite solution at 1.0 mL to the tubes in accordance with the order of the sequence and allowed them to stand for 6 min. Then, we added 10% aluminium nitrate solution at 1.0 mL to the tubes and let them stand for 6 min. After that, we added 4% sodium hydroxide solution at 6 mL and 70% ethanol scaled up to 20 mL. The mixture was mixed well and allowed them to stand for 15 min. Test tube No. 0 was used as a blank control. The mixture measured the absorbance at 510 nm. Then, the linear regression was plotted with concentration (Y) and absorbance (X).

### 2.7. Determination of Diosmetin-7-O-rutinoside (Diosmin) Content of Catalpa bungei C. A. Mey Callus

Sample preparation: The calluses were oven-dried at 60 °C, ground into powder and passed through a 60-mesh sieve. The 0.2 g powders were weighed into a conical flask; we added 30 mL of methanol and subjected them to ultrasonic extraction for 30 min. Then, we filtered the solution using a 0.22 μm filter membrane.

Preparation of standard solution: A total of 20 mg of geranylgeranyl standard powder was weighed precisely. A small amount of dimethyl sulfoxide (DMSO) was added to be dissolved completely, and then methanol was added to 100 mL to make a 0.2 mg/mL geranylgeranyl standard solution.

HPLC (1200 Infinity Series, Agilent, Santa Clara, CA, USA) detection conditions: The column was Eclipse Plus C18 (250 mm × 4.6 mm, 5 μm). The mobile phases were methanol (A) and 0.1% formic acid aqueous solution (B), and the elution procedure was 0–15 min,

50% A and 50% B at a flow rate of 1 mL/min. The column temperature was 30 °C. The injection volume was 10 μL. The detection wavelength was 296 nm.

### 2.8. Statistical Analysis

SPSS 22.0 software was used for statistical analysis of the data. One-way analysis of variance (ANOVA) was used to analyze the different treatments, and then the Duncan test was used to make multiple comparisons. The significance level was set as a = 0.05. All data were expressed as mean ± standard error (SE). Unsupervised PCA (principal component analysis) was performed using statistics function prcomp within R (www.r-project.org) accessed on 31 August 2022. The data were unit-variance scaled before undergoing unsupervised PCA. SigmaPlot 10.0 software was used for drawing.

## 3. Results

### 3.1. Effect of Sterilization Treatments on the In Vitro Culture of Young Leaves of Catalpa bungei C. A. Mey

Young leaves were sterilized using six sets of disinfection treatments. From Table 1, it can be seen that leaves had zero contamination in groups 4, 5 and 6, but that browning and survival rates were higher in groups 5 and 6 than in 4. Group 4 treatment (75% alcohol disinfection for 30 s and 0.1% mercuric ascension disinfection for 4 min) had the best disinfection effect, with a survival rate of 83.33%. This was significantly different from that of the other treatment groups.

### 3.2. Effect of Culture Medium on the Induction of Leaf Callus and Flavonoids Content

The sterilized leaves were inoculated into the prepared induction medium in a flat manner and the leaves first became larger and then the calluses grew along the edge of the leaves. In terms of callus tissue morphology, the groups with basic media of MS and DKW had more viable yellow-green-colored callus tissues than the group with basic media of N6 (Figure 2). As shown in Table 2, the highest rate of callus induction was observed in group H with 69.45%. Next, the callus induction rate of groups B, C, G and I also reached more than 60%. The results of polar analysis of variance (ANOVA) showed that the magnitude of the effect of these three factors on the induction of callus in *C. bungei* C. A. Mey was as follows: basic medium > 6-BA concentration > NAA concentration. By comparing the magnitude of K value, it could be seen that the optimum medium was DKW, the optimum 6-BA concentration was 2.0 mg·L$^{-1}$, and the optimum NAA concentration was 1.0 mg·L$^{-1}$. From the analysis of variance in shown Table 3, it can be seen that the effect of NAA concentration on the induction of callus was not significant.

**Table 3.** ANOVA results based on callus rate.

| Source | Type III Sum of Squares | Degree of Freedom | Mean Square | F | Sig. |
|---|---|---|---|---|---|
| Basic medium | 21,590.012 | 2 | 10,795.006 | 285.417 | 0.000 |
| 6-BA | 324.074 | 2 | 162.037 | 4.284 | 0.028 |
| NAA | 15.438 | 2 | 7.719 | 0.204 | 0.817 |
| Error | 756.438 | 20 | 37.822 | | |

In addition, the total flavonoid content was analyzed on the basis of each group. As shown in Table 2, the content of group G was significantly higher than that of the other groups. The results of polar analysis of variance (ANOVA) showed that the magnitude of the impact of these three factors on the total flavonoid content of *C. bungei* C. A. Mey leaf callus was as follows: basic medium > 6-BA concentration > NAA concentration. Comparing the magnitude of K value, it could be seen that the optimum medium was DKW, the optimum concentration of 6-BA was 2.0 mg·L$^{-1}$, and the optimum concentration of NAA was 0.5 mg·L$^{-1}$. From the analysis of variance in Table 4, it could be seen that the concentration of NAA had a non-significant effect on the total flavonoids content of the

callus. In conclusion, the suitable induction medium for leaf callus was DKW + 2.0 mg·L$^{-1}$ 6-BA + 0.5–1.0 mg·L$^{-1}$ NAA. The leaves became larger and thicker, and a small amount of callus tissue grew on the edge of the leaves after 7 days of inoculation (Figure 2j). The callus tissue increased in size significantly after 14 days of inoculation (Figure 2k), and callus tissue growth stabilized after 28 days of inoculation (Figure 2l).

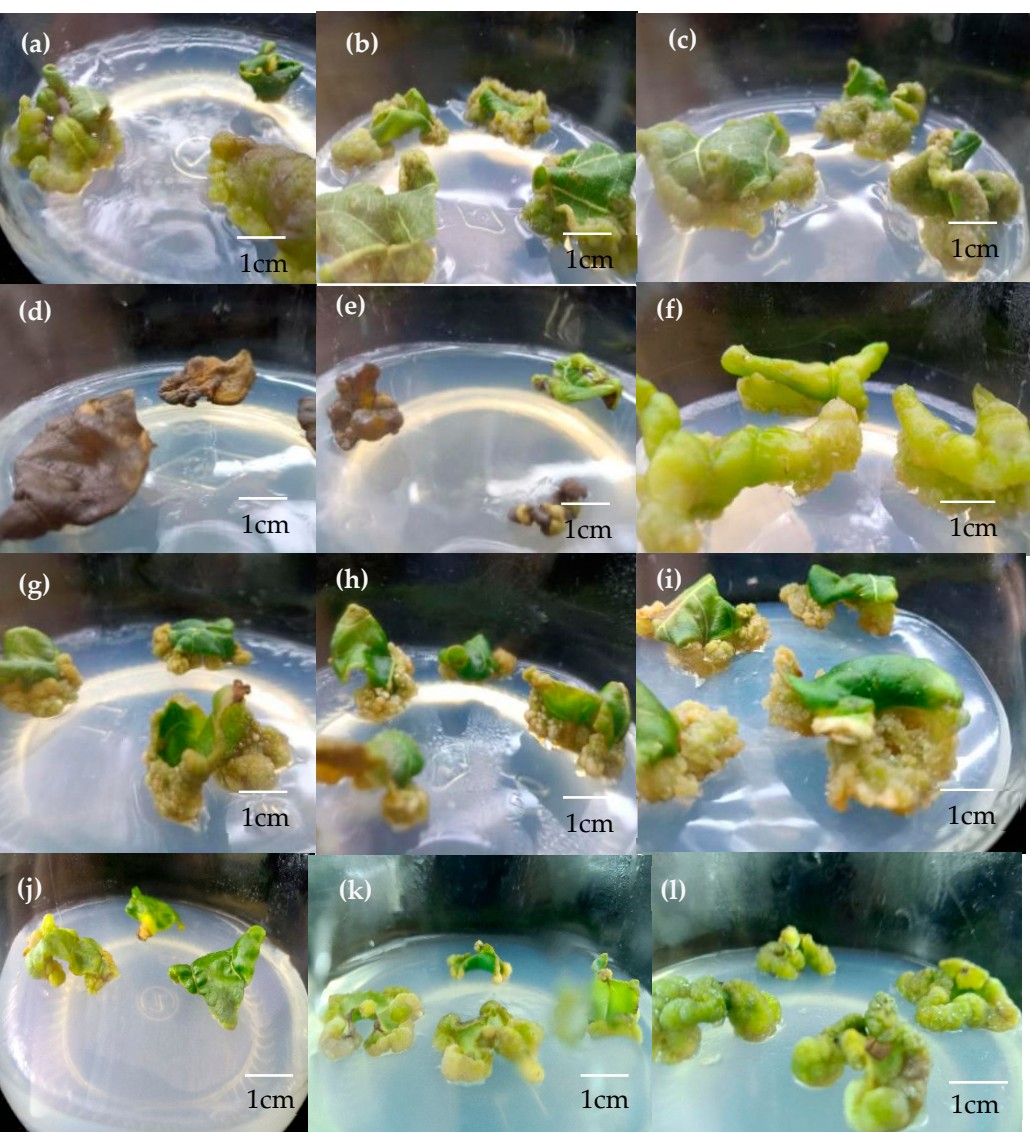

**Figure 2.** Growth status of callus: (**a**) the callus grown on the group A medium; (**b**) the callus grown on the group B medium; (**c**) the callus grown on the group C medium; (**d**) the callus grown on the group D medium; (**e**) the callus grown on the group E medium; (**f**) the callus grown on the group F medium; (**g**) the callus grown on the group G medium; (**h**) the callus grown on the group H medium; (**i**) the callus grown on the group I medium; (**j**) the callus grown for 7 days on the optimal induction medium; (**k**) the callus grown for 14 days on the optimal induction medium; (**l**) the callus grown for 28 days on the optimal induction medium.

**Table 4.** ANOVA results based on total flavonoid content.

| Source | Type III Sum of Squares | Degree of Freedom | Mean Square | F | Sig. |
|---|---|---|---|---|---|
| Basic medium | 32,371.802 | 2 | 16,185.901 | 101.729 | 0.000 |
| 6-BA | 1872.123 | 2 | 936.061 | 5.883 | 0.010 |
| NAA | 271.954 | 2 | 135.977 | 0.855 | 0.440 |
| Error | 3182.152 | 20 | 159.108 | | |

*3.3. Quality Control of Extensively Targeted Metabolomic Samples of Catalpa bungei C. A. Mey Flavonoids*

All the samples were analyzed using principal component analysis (PCA) method, and the results showed (Figure 3a) that the contribution rate of PC1 was 49.62% and that of PC2 was 33.25%, with a clear trend of separation between the samples, suggesting that there were significant differences among the samples. In this study, three groups of OPLS-DA models were established for ZP vs. YS, ZP vs. GP, and GP vs. YS, and the R2 and Q2 of the three groups of models were high (Figure 3b), meaning that the models had good predictive ability and reliability. The results of cluster analysis showed (Figure 3c) that the metabolic profiles of ZP, GP, and YS differed greatly. ZP, GP, and YS were higher in regions I, II, and III, respectively.

**(a)**                **(b)**                **(c)**

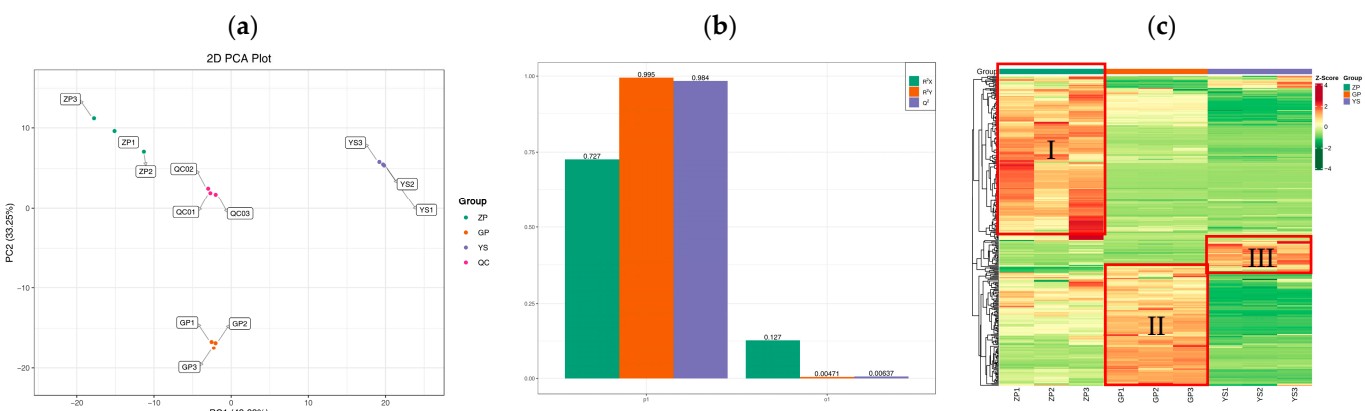

**Figure 3.** Overall characteristics of metabolites in leaves and callus of *C. bungei* C. A. Mey: (**a**) PCA score plot; (**b**) model validation plot; and (**c**) metabolite clustering heat map.

*3.4. Differences in Flavonoid Metabolome of Different Materials of Catalpa bungei C. A. Mey*

Differential metabolite screening was shown in Figure 3. In total, 338, 339, and 331 metabolites were identified in ZP, GP, and YS, respectively. Overall, 247, 182, and 226 differential metabolites were screened in the three groups of ZP vs. YS, ZP vs. GP, and GP vs. YS, respectively. A total of 247 differential metabolites were contained in the ZP vs. YS group, of which 24 were up-regulated and 223 were down-regulated (Figure 4a). The ZP vs. GP group contained 182 differential metabolites, of which 53 were up-regulated and 129 were down-regulated (Figure 4b). The GP vs. YS group contained 226 differential metabolites, of which 47 were up-regulated and 179 were down-regulated (Figure 4c). This demonstrated that flavonoids were further diversified in *C. bungei* C. A. Mey leaves during the induction of callus.

The multiplicity of different metabolites among the 3 groups was log2 processed to plot the multiplicity of difference, and the metabolites with top-20 multiplicity of difference values were listed (Figure 5). The top 20 metabolites in the ZP vs. YS group included 9 flavonoids, 5 flavonols, 2 isoflavonoids, 3 chalcones, and 1 flavanol (Figure 5a). The top 20 metabolites in the ZP vs. GP group included 8 flavonoids, 6 flavonols, 3 dihydroflavonoids, 2 isoflavonoids, and 1 flavanol (Figure 5b). The top 20 metabolites in the GP vs. YS group included 6 flavonoids, 7 flavonols, 1 isoflavonoid, 4 chalcones, 1 dihydroflavonol, and 1 flavanol (Figure 5c). In order to study the trend of relative content change of flavonoid

metabolites in different subgroups, the mean of relative content of differential metabolites in each group was standardized using z-score and then subjected to K-means (K-means) cluster analysis (Figure 6). The results showed a decreasing trend in the content of the vast majority of metabolites from ZP, GP to YS. Comparing the wrinkled-barked tree leaves with the light-barked tree leaves, the greatest changes in content came in terms of flavonoids and flavonol metabolites. The content of lignans (high in ZP) and their glycoside lignans (high in GP) differed greatly between the two types of leaves, which was presumed to be related to the growth differences between the two types of leaves. The content of 30 (Sub Class 6) metabolites, including flavonoids (19), flavonols (9), dihydroflavonoids (1), and dihydroflavonols (1), was significantly increased in calluses as compared with leaves. It was possible for isoflavonoid and chalcone metabolites to be transformed or consumed during cellular secondary metabolism during the process of callus induction in leaves.

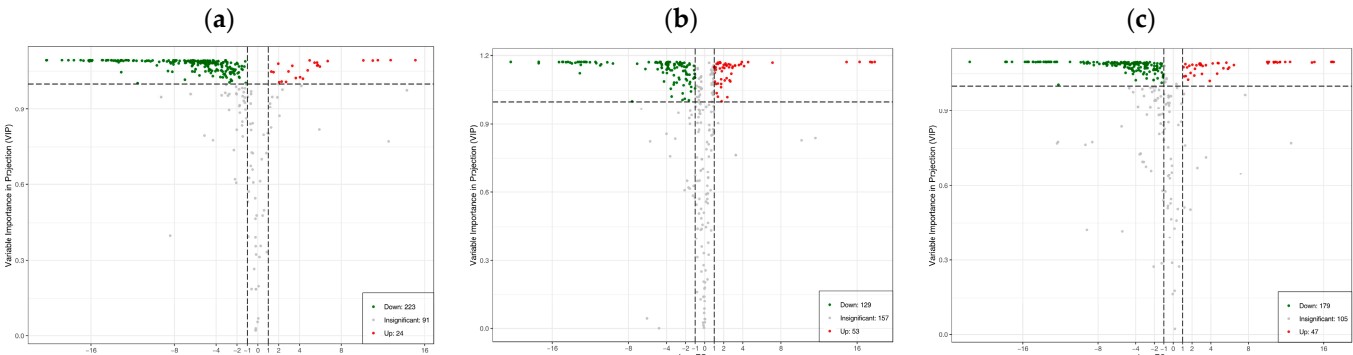

**Figure 4.** Volcanic maps of differential metabolites: (**a**) ZP vs. YS group; (**b**) ZP vs. GP group; and (**c**) GP vs. YS group.

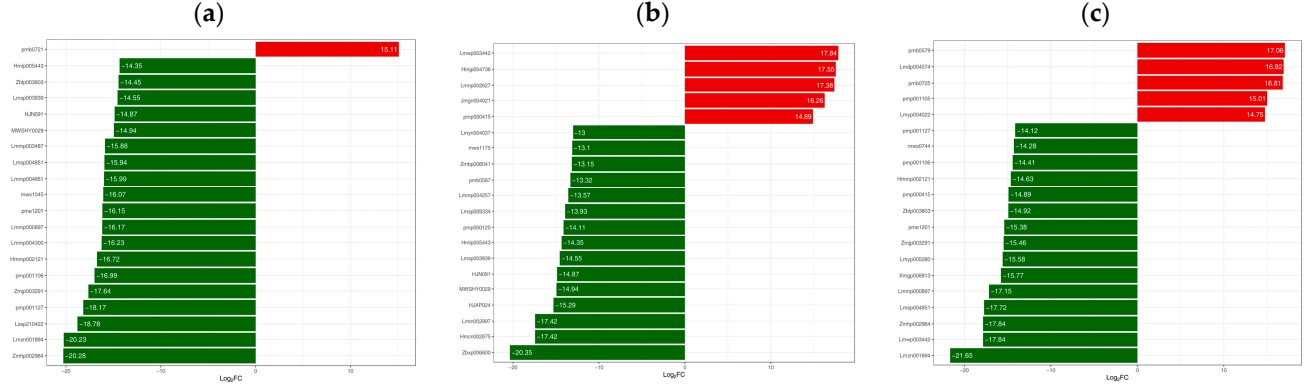

**Figure 5.** Differential metabolites Top 20 metabolites: (**a**) ZP vs. YS group; (**b**) ZP vs. GP group; and (**c**) GP vs. YS group.

In order to understand the differences in the composition of flavonoid metabolites between the callus and the two leaf types, differential metabolites between the groups were screened using t-tests ($p < 0.05$) and the OPLS-DA model (VIP > 1). As shown in Figure 7a, there were 98 total differential metabolites between the three samples. As shown in Figure 7b,c, there were 37 flavonoid metabolites with increased relative content in the calluses compared with the light-barked *C. bungei* C. A. Mey leaves, and there were 21 flavonoid metabolites with increased relative content in the calluses compared with the wrinkled-barked *C. bungei* C. A. Mey leaves. After comparative analysis, there were eight flavonoid metabolites whose relative contents were increased in the calluses in both groups (Table 5). Among them, diosmin had the shortest retention time for efficient detection and was prominent for its medicinal functions in the treatment of various vascular diseases, including varicose veins, poor circulation in the legs (venous stagnation) and bleeding from the eyes or gums. Diosmin had also been used to prevent hepatotoxicity.

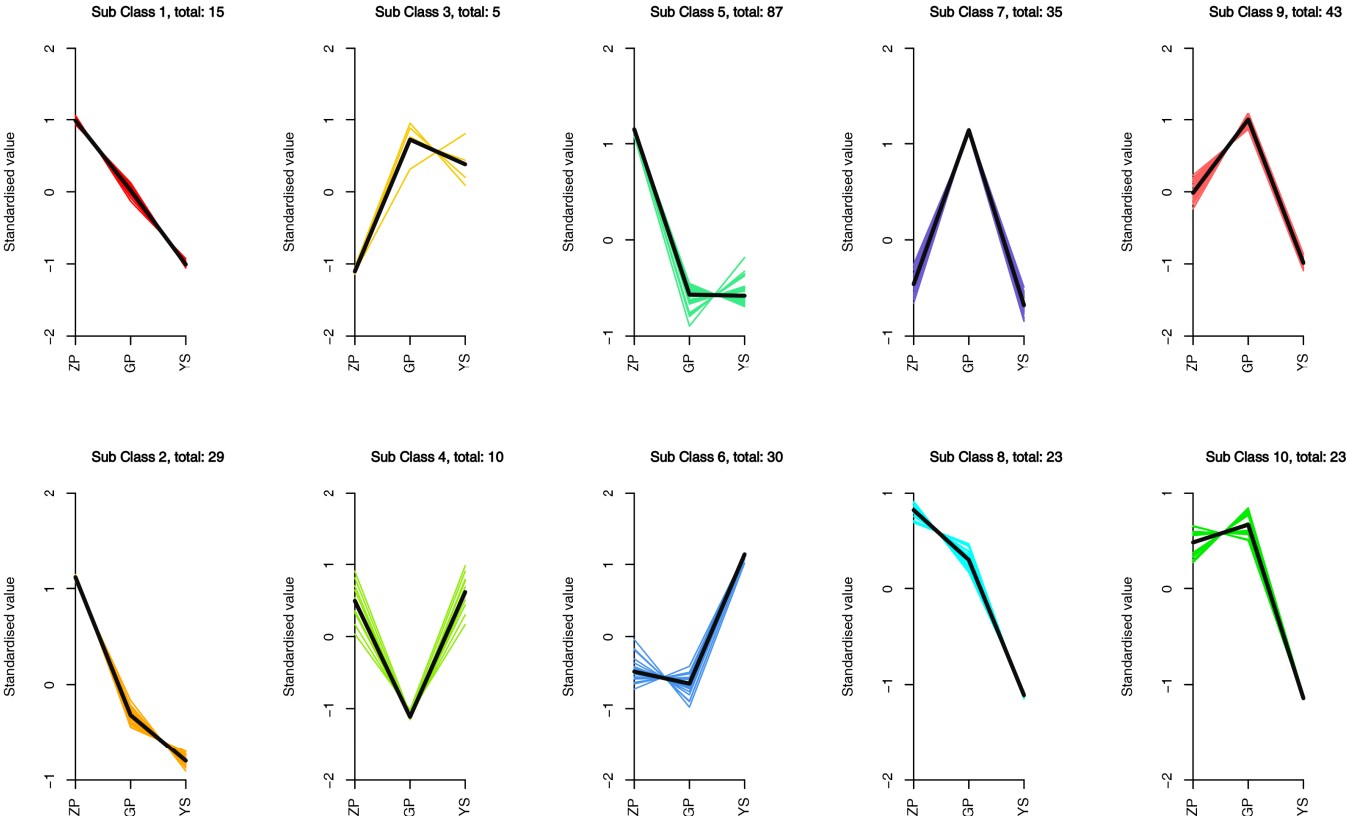

**Figure 6.** K-means clustering analysis. The horizontal axis represented the grouping of samples, the vertical axis represented the standardized relative content of metabolites, Sub Class represented the category number of metabolites with the same trend of change, and total represented the number of metabolites in that category.

(**a**)　　　　　　　　　　(**b**)　　　　　　　　　　(**c**)

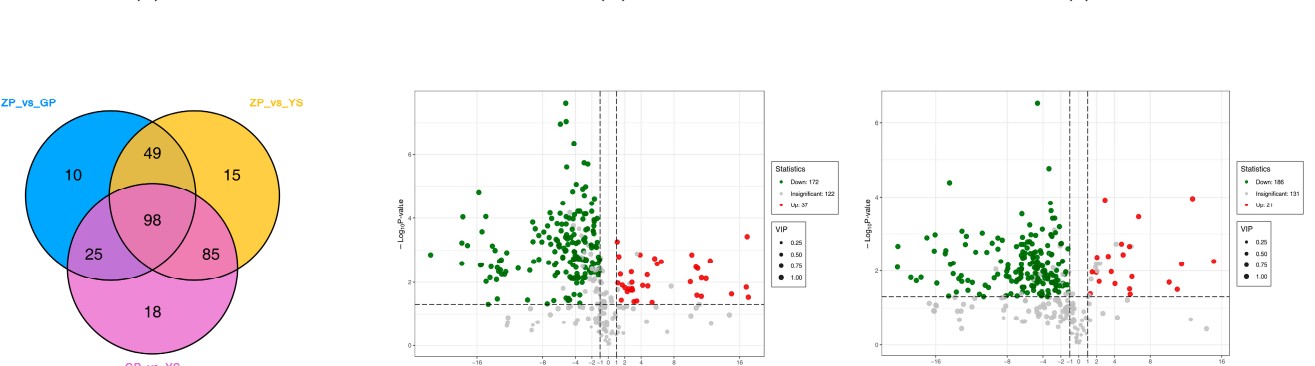

**Figure 7.** Screening of differential metabolites of flavonoids: (**a**); Venn diagram analysis of metabolites in ZP vs. YS group, ZP vs. GP group and GP vs. YS group; (**b**) volcanic map of differential metabolites in GP vs. YS group; (**c**) volcanic map of differential metabolites in ZP vs. YS group. Each dot represented a metabolite. Green dots in the figure represented differentially expressed metabolites with down-regulated relative content, red dots represented differentially expressed metabolites with up-regulated relative content, and grey dots represented metabolites that were detected but had insignificant differences in relative content. Numbers in the legend represented the number of individuals.

**Table 5.** Major differential metabolites.

| Metabolite | CAS | ZP vs. YS | | | GP vs. YS | | |
|---|---|---|---|---|---|---|---|
| | | VIP | *p*-Value | Log$_2$FC | VIP | *p*-Value | Log$_2$FC |
| Diosmetin-7-O-rutinoside (diosmin) | 520-27-4 | 1.05 | 0.04 | 1.29 | 1.07 | 0.02 | 2.31 |
| Tricin-7-O-(2″-feruloyl)glucoside | -- | 1.07 | 0.01 | 5.95 | 1.10 | 0.01 | 16.81 |
| Kaempferol-3-O-(2″-galloyl)glucoside | 76343-90-3 | 1.01 | 0.14 | 2.71 | 1.09 | 0.11 | 14.75 |
| Quercetin-3-O-(2‴-O-feruloyl)sophoroside | -- | 1.09 | 0.00 | 6.70 | 1.10 | 0.00 | 16.92 |
| Chrysoeriol-7-O-(6″-sinapoyl)glucoside | -- | 1.07 | 0.03 | 5.69 | 1.09 | 0.03 | 17.06 |
| Kaempferol-3-O-neohesperidoside-7-O-glucoside | -- | 1.03 | 0.02 | 4.05 | 1.09 | 0.02 | 15.01 |
| Quercetin-3-O-(2‴-O-caffeoyl)sophoroside | -- | 1.07 | 0.00 | 3.30 | 1.09 | 0.01 | 4.74 |
| Quercetin-3-O-(6″-O-Caffeoyl)sophoroside-7-O-rhamnoside | -- | 1.05 | 0.00 | 2.94 | 1.09 | 0.00 | 5.65 |

*3.5. Effect of Elicitors on the Induction of Callus in Catalpa bungei C. A. Mey*

According to the result of 3.2, the suitable induction mediums for leaf callus tissue were named J (DKW + 2.0 mg·L$^{-1}$ 6-BA + 1.0 mg·L$^{-1}$ NAA) and K (DKW + 2.0 mg·L$^{-1}$ 6-BA + 0.5 mg·L$^{-1}$ NAA), respectively.

As shown in Table 6, the induction rates of groups J and K without the addition of elicitor medium were 71.11% and 65.56%, respectively. There was a 100% induction rate in the J-SA1 group with 10 µmol·L$^{-1}$ SA added to medium J. Secondly, the induction rate was also significantly increased in the K-SA1 group with 10 µmol·L$^{-1}$ SA added to medium J, and the induction rate was 92.22%. This indicated that 10 µmol·L$^{-1}$ SA had a significant stimulative effect on the induction of leaf callus. When the concentration of SA was increased to 50 µmol·L$^{-1}$, the induction rate tended to decrease, and the addition of 100 µmol·L$^{-1}$ SA even had an inhibitory effect on the induction of calluses. The addition of YE to media J and K did not have any significant effect on the callus induction rate, whereas 200 mg·L$^{-1}$ of YE had an inhibitory effect on callus induction.

**Table 6.** Callus induction rates.

| Group | Hormone Concentration (mg·L$^{-1}$) | | Basic Medium | SA (µmol·L$^{-1}$) | YE (mg·L$^{-1}$) | Induction Rate (%) |
|---|---|---|---|---|---|---|
| | 6-BA | NAA | | | | |
| J | 2.0 | 1.0 | DKW | 0 | 0 | 71.11 ± 2.22 d |
| J-SA1 | 2.0 | 1.0 | DKW | 10 | -- | 100.00 ± 0.00 a |
| J-SA2 | 2.0 | 1.0 | DKW | 50 | -- | 52.22 ± 1.11 e |
| J-SA3 | 2.0 | 1.0 | DKW | 100 | -- | 52.22 ± 2.94 e |
| J-YE1 | 2.0 | 1.0 | DKW | -- | 50 | 44.44 ± 1.11 f |
| J-YE2 | 2.0 | 1.0 | DKW | -- | 100 | 77.78 ± 1.11 c |
| J-YE3 | 2.0 | 1.0 | DKW | -- | 200 | 36.67 ± 1.93 g |
| K | 2.0 | 0.5 | DKW | 0 | 0 | 65.56 ± 1.11 d |
| K-SA1 | 2.0 | 0.5 | DKW | 10 | -- | 92.22 ± 2.94 b |
| K-SA2 | 2.0 | 0.5 | DKW | 50 | -- | 76.67 ± 1.93 c |
| K-SA3 | 2.0 | 0.5 | DKW | 100 | -- | 50.00 ± 1.92 e |
| K-YE1 | 2.0 | 0.5 | DKW | -- | 50 | 67.78 ± 1.11 d |
| K-YE2 | 2.0 | 0.5 | DKW | -- | 100 | 34.44 ± 2.94 g |
| K-YE3 | 2.0 | 0.5 | DKW | -- | 200 | 37.78 ± 1.11 g |

Means followed by the same letters in rows are not significantly different at $p \leq 0.05$.

As shown in Table 7, the addition of both SA and YE increased the content of total flavonoids in the callus compared with the medium without the elicitors and the total flavonoid contents of the combinations J-YE2 and K-YE2 with the addition of 100 mg·L$^{-1}$ YE were significantly higher than those of the control group, being 168.652 mg.g$^{-1}$ and 161.392 mg.g$^{-1}$, respectively. Additionally, the content of total flavonoids in the callus increased by 41.65% and 59.20%, respectively, compared with the control group. The addition of YE had an increasing effect on the content of diosmin in the callus compared with that of the medium without the addition of the elicitors, whereas the addition of SA could not promote an increase in diosmin content. On the contrary, it had an inhibitory effect.

**Table 7.** Content of active components of calluses in different treatment groups.

| Group | Hormone Concentration (mg·L$^{-1}$) | | Basic Medium | SA (µmol·L$^{-1}$) | YE (mg·L$^{-1}$) | Total Flavonoid Content (mg.g$^{-1}$) | Diosmetin-7-O-rutinoside (Diosmin) Content (mg.g$^{-1}$) |
|---|---|---|---|---|---|---|---|
| | 6-BA | NAA | | | | | |
| J | 2.0 | 1.0 | DKW | 0 | 0 | 119.064 ± 1.062 i | 6.090 ± 0.903 bcd |
| J-SA1 | 2.0 | 1.0 | DKW | 10 | -- | 133.760 ± 1.116 g | 4.380 ± 0.006 de |
| J-SA2 | 2.0 | 1.0 | DKW | 50 | -- | 127.732 ± 3.01 gh | 4.440 ± 0.592 de |
| J-SA3 | 2.0 | 1.0 | DKW | 100 | -- | 143.308 ± 0.754 de | 5.505 ± 0.551 cde |
| J-YE1 | 2.0 | 1.0 | DKW | -- | 50 | 146.256 ± 1.284 d | 6.660 ± 0.195 bc |
| J-YE2 | 2.0 | 1.0 | DKW | -- | 100 | 168.652 ± 0.781 a | 7.590 ± 0.561 ab |
| J-YE3 | 2.0 | 1.0 | DKW | -- | 200 | 150.084 ± 1.010 c | 8.595 ± 0.602 a |
| K | 2.0 | 0.5 | DKW | 0 | 0 | 101.376 ± 1.00 j | 7.275 ± 0.283 abc |
| K-SA1 | 2.0 | 0.5 | DKW | 10 | -- | 140.272 ± 2.017 e | 6.840 ± 0.588 bc |
| K-SA2 | 2.0 | 0.5 | DKW | 50 | -- | 137.060 ± 2.026 f | 4.890 ± 0.208 de |
| K-SA3 | 2.0 | 0.5 | DKW | 100 | -- | 139.832 ± 1.260 f | 4.200 ± 0.563 e |
| K-YE1 | 2.0 | 0.5 | DKW | -- | 50 | 131.252 ± 1.010 g | 4.860 ± 0.928 de |
| K-YE2 | 2.0 | 0.5 | DKW | -- | 100 | 161.392 ± 0.746 b | 5.850 ± 0.190 cde |
| K-YE3 | 2.0 | 0.5 | DKW | -- | 200 | 126.632 ± 2.082 h | 6.945 ± 0.053 bc |

Means followed by the same letters in rows are not significantly different at $p \leq 0.05$.

## 4. Discussion

*Catalpa bungei*, belonging to the Bignoniaceae family, is a tall deciduous tree that has been cultivated for more than two thousand years [11]. *C. bungei* C.A. Meyer, originating in China, is a valuable timber species and a famous ornamental tree extensively found in the Yellow River and the Yangtze River basins. *C. bungei* is often used as a street tree due to its wide adaptability, strong environmental tolerance, and high ornamental value. This timber confers good physical and chemical properties, as well as good medicinal properties [8].

Widely targeted metabolomics analysis is a new method that combines the advantages of targeted and untargeted metabolomic technologies. The method utilizes Q TRAP mass spectrometry based on the multi-reaction monitoring (MRM) mode, allowing the simultaneous quantification of hundreds to thousands of metabolites with both breadth and high sensitivity [12]. This method has been applied in recent years for the detection and identification of active ingredients in plants [13–15]. Xu et al. [1] obtained two flavonoids named Luteolin and Apigenin with antioxidant activity via column chromatography and a biological activity tracking method. This study used widely targeted metabolomics techniques to detect 340 flavonoid metabolites from callus-extracted substances. There were 8 flavonoid metabolites in callus tissue that were significantly higher than those in leaves. Although Luteolin and Apigenin were not included, the result indicated that, under suitable culture conditions, a flavonoid content higher than that of natural plants can be obtained by inducing a callus of *C. bungei* C. A. Mey in a culture medium.

Some earlier studies showed, for example, that chandelin in the chandelier flower of the heartwort [16] and carotenoids in the calendula [17] were all important plant secondary metabolites, all of which could be obtained from isolated callus culture. The metabolomic results of the present study showed that, among the differential metabolites shared by the calluses and leaves, the flavonoid metabolite with the highest content and the shortest

retention time in the callus was diosmin. Diosmin has good biological activities such as hypoglycemic [18], antihypertensive [19], cardioprotective [20], anticoagulant and antivenous thrombosis [21,22], antitumor effects [23,24]. It is indicated that *C. bungei* C. A. Mey tissue culture technology can be used to induce callus, and that appropriate measures can be taken to enhance the accumulation of active ingredients, especially to increase the content of diosmin.

Studies showed that the accumulation of secondary metabolites in plants could be increased by optimizing cell culture conditions, feeding precursors and adding elicitors [25–27]. The use of elicitors to regulate the synthesis and accumulation of plant secondary metabolites is currently one of the most commonly used methods in plant cell culture. Wang et al. [28] significantly increased the content of paclitaxel to 1.58 mg·$g^{-1}$ by adding 20 mg·$L^{-1}$ SA (salicylic acid) during suspension culture of *Taxus chinensis* (Pilger) Rehd. cells. Ali et al. [29] treated *Artemisia absinthium* suspension cultures with methyl jasmonate and jasmonic acid, which increased the content of phenolic and flavonoid constituents. In addition, using an appropriate amount of sodium salicylate, chitosan, methyl jasmonate, and ascorbic acid also promoted the accumulation of platyphylline in the callus of *Platycodon grandiflorus* [30].

The growth status of the callus varied greatly among different elicitor treatments. The results of this study showed that the color of salicylic acid-treated *C. bungei* C. A. Mey callus was yellowish, and the calluses were more prone to browning during growth as the concentration of SA increased (10 μmol·$L^{-1}$–100 μmol·$L^{-1}$). This was similar to the induction results of SA when added to a ginger callus culture. Salicylic acid promoted or inhibited the growth in ginger calluses depending on its concentration. The growth in callus and flavonoid contents could be stimulated when the concentration of salicylic acid was appropriate, but the growth in the callus and flavonoid contents were inhibited when the concentration of salicylic acid exceeded the threshold for normal growth in the callus [31]. The YE (yeast extract)-treated callus grew well, showing a healthy light green color in the culture plates. As the concentration of YE increased (50 mg·$L^{-1}$–200 mg·$L^{-1}$), the color gradually turned yellowish. Although the rate of callus induction was not as high as that of the experimental group with the addition of SA, culture conditions with the addition of 100 mg·$L^{-1}$ YE could effectively increase the content of diosmin and total flavonoids in the callus of *C.* C. A. Mey, which was comparable to that of the YE-treated *Swertia chirata* callus, albeit with significantly higher flavonoid contents [32]. This might be due to the fact that yeast extract can achieve the purpose of increasing the production of specific secondary metabolites by activating the activity of secondary metabolite synthase [33]. In future studies, we intend to research the suspension culture system of *C. bungei* C. A. Mey cells with the addition of elicitors based on the results from this study, aiming to produce medicinal metabolites from plants in large quantities.

## 5. Conclusions

The young leaves of *C. bungei* C. A. Mey were effectively sterilized with 75% alcohol for 30 s and 0.1% mercuric acid for 4 min to achieve the best disinfection effect. The basic medium type and 6-BA concentration have significant effects on the growth and flavonoid content of calluses, while NAA concentration has no effect on them. With the yhe young leaves of *C. bungei* C. A. Mey as the material, the best callus induction medium was DKW + 2.0 mg·$L^{-1}$ 6-BA + 0.5–1.0 mg·$L^{-1}$ NAA. The rate of callus induction was also affected by SA, and the highest callus induction rate was achieved with the addition of 10 μmol·$L^{-1}$ SA to DKW + 2.0 mg·$L^{-1}$ 6-BA + 1.0 mg·$L^{-1}$ NAA + 30 g·$L^{-1}$ sucrose + 6.8 g·$L^{-1}$ agar medium. Flavonoid metabolite abundance in *C. bungei* C. A. Mey callus was significantly higher than that in *C. bungei* C. A. Mey leaves as analyzed using the flavonoid broadly targeted metabolomic technique. The flavonoid metabolite that was significantly more abundant and widely induced in the callus of *C. bungei* C. A. Mey leaves was identified as diosmin. YE had a greater effect on the content of flavonoid in the callus, and the addition of 100 mg·$L^{-1}$ YE to DKW + 2.0 mg·$L^{-1}$ 6-BA + 1.0 mg·$L^{-1}$ NAA + 30 g·$L^{-1}$ sucrose + 6.8 g·$L^{-1}$ agar

medium resulted in the highest content of total flavonoids in the callus. With the addition of 200 mg·L$^{-1}$ YE, the highest content of diosmin was found in the callus.

**Author Contributions:** Y.Z. conceived and designed the experiments; J.H. provided the plant materials; X.Z. and X.W. performed the experiments, analyzed the data, prepared the figures and tables, and they contributed equally to this work; Z.L. reviewed drafts of the paper. All authors have read and agreed to the published version of the manuscript.

**Funding:** This research was funded by National Science and Technology Achievement Promotion Project of National Forestry and Grassland Administration (2020133123) and Hunan Province Forestry Science and Technology Innovation Project (XLKY202221).

**Data Availability Statement:** All data are available in the manuscript.

**Conflicts of Interest:** The authors declare no conflict of interest.

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
