# Peer review of "Flavonoid Metabolome-Based Active Ingredient Mining and Callus Induction in Catalpa bungei C. A. Mey"

_forests, doi:10.3390/f14091814_

Round 1
Reviewer 1 Report
Dear Authors
There are some literary and design mistakes. Please visit reviewed file. Some most important mistakes: 1. Some sentences should be revised. 2. Abbreviation technique has not been used properly. 3. Introduction section is lack of background. 4. Explanation about medicinal properties of the species is too. 5. MM section is confusing. 6. Order of figures in the text is completely mistake. 7. There is no space between some words and signs.

Reviewer 2 Report
Dear Authors,
The manuscript with the title «Flavonoid metabolome-based active ingredient mining and callus induction in Catalpa bungei C. A. Mey» is devoted to obtaining and analyzing calluses of Catalpa bungei for flavonoid content. An extensive study was carried out: a sterilization scheme was selected for plant material, the medium for callus initiation. It is shown that the flavonoids content were higher in callus tissue than in leaves. The results obtained in this study are of interest from both a fundamental and practical standpoint.
However, there are a number of fundamental comments to be made about the study. The age of the callus used in the experiments is not clear. The figure shows the formation of the callus on the explants. This type of callus is not suitable for the experiments. It is primary callus. Studies should use callus tissue, which has stable good growth. We would like to see growth curves for calluses in the studied environments, as well as images of growing calluses.
Reviewer 3 Report
In this manuscript, authors have reported research entitled “Flavonoid metabolome-based active ingredient mining and callus induction in Catalpa bungei C. A. Mey”. In general, the manuscript was satisfactorily well presented. However, some small issues need attention.
1- Improve the quality of the figures. For example, in Figure 2, the label can be increased to improve readability.
2- Check the English grammar and structure throughout the manuscripts, though overall it is well written.
3- Use more descriptive expressions in the conclusion part.
Check the English grammar and structure throughout the manuscripts, though overall it is well written.
Round 2
Reviewer 2 Report
Dear Authors,
In my opinion, the main issue with this article is that it analyses primary callus. However, it is well-known that primary callus is not a reliable culture. It is necessary to analyze tissue that has undergone several subculture periods. Biomass growth is an essential indicator absent from the article.
